# Quantitative Assessment of the Inadequate Intake of Macronutrients, Minerals, and Vitamins Associated with Ultra-Processed Food Consumption

**DOI:** 10.3390/ijerph21070888

**Published:** 2024-07-08

**Authors:** Raiane M. Costa, Antonio G. Oliveira, Karina G. Torres, Anissa M. Souza, Gabriela S. Pereira, Ingrid W. L. Bezerra

**Affiliations:** 1Postgraduate Program in Health Sciences, Centro de Ciências da Saúde, Universidade Federal do Rio Grande do Norte, Natal 59012-570, RN, Brazil; raiane.costa.092@ufrn.edu.br (R.M.C.); antonio.gouveia@ufrn.br (A.G.O.); karina.torres@ufrn.br (K.G.T.); anissa.souza.094@ufrn.edu.br (A.M.S.); gaabsp@gmail.com (G.S.P.); 2Pharmacy Department, Centro de Ciências da Saúde, Universidade Federal do Rio Grande do Norte, Natal 59012-570, RN, Brazil; 3Nutrition Department, Centro de Ciências da Saúde, Universidade Federal do Rio Grande do Norte, Natal 59078-970, RN, Brazil

**Keywords:** nutrient intake, ultra-processed food, food consumption, diet quality, dietary reference intakes, dietary inadequacies

## Abstract

Studies indicate that ultra-processed food (UP) consumption correlates negatively with essential vitamin and mineral intake and positively with sodium and lipid intake. The objective of this study was to explore the relationship between UP consumption and deviations from nutritional guidelines. An observational, cross-sectional analytical study was conducted on a probability sample of manufacturing workers in the state of Rio Grande do Norte, Brazil. Food consumption was assessed with a 24 h recall survey, and nutrient intake inadequacies were calculated as the difference between individuals’ intake of energy, macronutrients, minerals and vitamins, and the dietary reference intakes for individuals of the same sex and age group, and then analyzed for trends across the percentage contribution of UP to total energy intake with nonparametric multiple regression adjusted for covariates. The study included 921 workers from 33 industries, 55.9% male, with a mean age of 32 years. Overall, the study population exhibited deficits in energy, all macronutrients, and in some micronutrients. With increasing UP contribution to total energy intake, there is a trend towards a greater intake of energy (*p* < 0.001), total, saturated, monounsaturated, and trans fats (*p* < 0.001), n6-polyunsaturated fatty acids (*p* = 0.03), carbohydrates (*p* < 0.001), calcium (*p* = 0.008), and manganese (*p* < 0.001), thiamin (*p* < 0.001), and vitamin B6 (*p* = 0.01); however, this comes with a negative consequence in terms of reducing the protein consumption (*p* = 0.037), fiber (*p* = 0.035), copper (*p* = 0.033), and vitamin E (*p* = 0.002) intake. The results show that correcting energy and micronutrient deficiencies by increasing UP consumption can also lead to a decrease in diet quality.

## 1. Introduction

The composition of a diverse, balanced, and healthy diet varies according to personal characteristics, cultural context, locally available foods, and dietary habits. However, the fundamental principles of a healthy diet remain consistent across populations [1]. Nutritional recommendations aim to promote a balanced diet that fulfills the physiological needs of the majority [2,3].

Nevertheless, global lifestyle shifts, driven by rapid urbanization and increased processed food production, have reshaped dietary patterns, leading to altered nutrient intake and a surge in processed and ultra-processed (UP) food consumption [1,4,5]. UP items undergo extensive industrial processing, chemically or physically transforming them. These items, often packaged and shelf-ready in supermarkets, typically contain more than five ingredients and have long shelf lives. Enhanced with additives, flavors, stabilizers, emulsifiers, and colorants for heightened appeal, UPs are hyper-palatable and potentially addictive, influencing adherence to unhealthy eating patterns [6,7,8,9]. Busy schedules, irregular work hours, and a lack of commitment to healthy eating contribute to adults’ increased reliance on UPs [10,11]. With minimal culinary preparation required, UPs are increasingly prevalent in diets worldwide [12].

However, UPs are typically energy-dense, high in fats, refined starches, free sugars, and salt, while lacking in protein, dietary fiber, and micronutrients [13,14]. Studies indicate that greater UP consumption correlates negatively with essential vitamin and mineral intake and positively with sodium and lipid consumption [15,16,17]. Moreover, high UP intake is associated with low dietary diversity and micronutrient deficiency [18], as well as non-communicable diseases, like obesity [19], increased risk of mortality, cardiovascular diseases [20], and cancer [21].

Several studies that investigated the association between the consumption of UPs and the nutrient content of diets have consistently found a trend towards increased amounts of energy, carbohydrates, total and saturated fat in the composition of diets, with simultaneous decrease in protein and fiber, as well as a number of minerals and vitamins [22]. All those studies have reported only the average consumed amounts of nutrients across categories of UP consumption, with a smaller number also estimating the proportion of subjects consuming inadequate amounts of nutrients, with cut-off levels usually based on tables of estimated average requirements or tolerable upper intake levels. However, no study has yet, to the best of our knowledge, estimated the magnitude of the deviations from recommended dietary allowances associated with levels of UP consumption.

Therefore, to further contribute to the evaluation of the implications of UP consumption to the composition of diets, the aim of this study was to quantify the eventual deviations from nutritional guidelines in macronutrient, energy, mineral, and vitamin intake associated with UP consumption.

## 2. Materials and Methods

### 2.1. Study Design

This was an observational, cross-sectional, analytical study based on a combined stratified proportional and multistage population survey, conducted on a probability sample of manufacturing workers. The study was carried out in the state of Rio Grande do Norte, located in northeastern Brazil. The strata considered were the industry size, with 3 levels (<50, 50–500, and >500 workers), and the sector of activity, with 3 levels corresponding to sectors with the greatest economic representation in the state (food and beverages, non-metallic minerals, and textiles). The study protocol was approved by the Research Ethics Committee of the Onofre Lopes University Hospital under authorization number 2.198.545 of 2 August 2017. Prior to the study interviews, all subjects gave informed consent to participate in the study in writing, and the study was conducted according to the principles set forth in the Declaration of Helsinki, as revised in 2013.

### 2.2. Study Population

The first stage of the sampling plan consisted of manufacturing industries randomly selected in proportion to the total number of industries in the state for each combination of stratification factors, using the industries’ registry provided by the Federation of Industries of the State of Rio Grande do Norte as the sampling frame. All industries that agreed in writing to participate in the research were included. The second stage consisted of a sample of workers from each industry included in the first stage, with a fixed sample size, obtained by simple randomization from a list of all eligible workers provided by the human resources department of each industry. Eligible workers were those over 18 years of age, with a formal employment relationship with the company, who regularly used the company cafeteria for lunch, and who provided written consent to participate in the research. Pregnant workers, temporary employees, and interns were excluded.

### 2.3. Study Plan

Data collection occurred in person from Tuesdays to Saturdays to document the habitual food consumption of workers. Depending on the company’s size, one to four visits were made, utilizing spaces provided by the companies.

Demographic data including age, sex, education level, marital status, number of children, monthly income, and participation in internal professional training activities were collected. Dietary intake information was gathered through the 24 h dietary recall (24HR) method, employing the multiple pass methods methodology to minimize errors. This method assists the interviewee in recalling foods consumed the day before the interview by breaking down the process into stages, such as describing the foods consumed, mealtimes, quantities in household measures, and preparation methods, among others [23].

The dietary intake data obtained from the 24HR method were initially quantified by converting household measures reported by interviewees into metric units of weight and volume using established references (direct weighing of foods, photographic records, and specific manuals) [24]. Subsequent nutritional analyses were conducted based on the dietary intake information, utilizing centesimal composition tables of available foods [25,26,27] or information from food labels if analysis was unavailable.

Foods recorded in the 24HR survey were classified according to the NOVA classification system, which categorizes foods based on the type, extent, and purpose of industrial processing they undergo. The classes are as follows: 1—unprocessed or minimally processed (acquired for consumption without undergoing alterations or subjected to minimal changes after leaving nature); 2—culinary ingredients (products extracted from unprocessed foods and used for seasoning and cooking, such as salt, sugar, oils, and vinegar); 3—processed foods (essentially manufactured with the addition of culinary ingredients, for example, salt or sugar added to unprocessed or minimally processed foods); and 4—ultra-processed foods (products whose production involves several stages of processing and various ingredients, many of which are exclusively industrial) [12,28]. In homemade preparations from R24H containing items from various processing classes, all ingredients were dissected according to quantities based on preparation technical sheets. The quintiles of the percentage contribution of UPs to total energy consumption were then determined.

The calculation of nutritional inadequacies commenced with the estimation of daily caloric needs based on Estimated Energy Requirement (EER) equations, which aim to maintain energy balance in healthy individuals. These equations consider age, sex, and level of physical activity [29,30]. The level of physical activity was determined using a validated translation of the short version of the International Physical Activity Questionnaire (IPAQ) into Portuguese [31,32]. The IPAQ questions about three types of activity: walking, moderate-intensity activities, and vigorous-intensity activities related to physical activities in leisure, household, gardening, work, and transportation mentioned by the respondent, and estimates the metabolic equivalent of the task per minute per week (MET.min/week) from the amount of self-reported time spent in the previous 7 days on each type of physical activity. According to the participant’s responses, their level of physical activity was classified into low, moderate, and high [31,32].

For the calculation of inadequacies, reported intakes of each macronutrient, mineral, and vitamin were compared with internationally adopted reference values from several sources, depending on the nutrient [29,33,34,35,36]. Nutrient intake inadequacies were calculated as the difference between individual intake and the recommended value for individuals of the same sex and age group, divided by the recommended value and expressed as a percentage.

### 2.4. Statistical Analysis

The subjects’ characteristics are presented for the whole sample and by sex with summary statistics of the mean and standard deviation, or absolute and relative frequencies. The inadequacies in nutrient intake, defined as the percent deviations from the recommended allowances, were computed as the difference between actual intake and recommended allowance, divided by recommended allowance × 100, are presented descriptively as the means of percentage deviation of each daily nutrient intake relative to dietary recommendations, considering individual characteristics (which may include sex, age group, and level of physical activity), across the quintiles of the percentage contribution of UP to total energy intake, as well as for the whole study sample. For the investigation of the association of UP consumption with nutrient inadequacy, a search for potential confounders was performed with stepwise multiple linear regression of the sociodemographic variables (age, sex, marriage status, number of children, education, income, in-house training, and participation in a food assistance program) on the percent contribution of UP consumption to total energy intake. The variables selected by backward elimination (*p* < 0.05) were included as covariates in all statistical analyses. Because of the possibility of non-linear associations between UP consumption and inadequate nutrient intake, as well as the highly skewed distribution of most nutrients, nonparametric multiple regression was used to test the association of nutrient intake with UP consumption. Stata 15.1 (Stata Corporation, College Station, TX, USA) was used for the statistical analysis, and a *p*-value < 0.05 was considered to be evidence of statistical significance.

## 3. Results

The study was conducted between September 2017 and July 2018 in 33 industries, divided into small-sized (*n* = 13), medium-sized (*n* = 14), and large-sized (*n* = 6). This included 13 from the *textile* sector, 6 from the non-metallic minerals sector, and 14 from the food and beverage sector. A total of 921 workers participated in the study (406 women and 515 men), with no refusals to participate.

The study population had an average age of 38 years, with 62.7% being married or cohabiting. Half of the sample had 1 to 2 children, and nearly half had completed high school (Table 1). On average, they earned 1.46 times the minimum wage, and only 18.7% had specific training for their industry position. When broken down by sex, women averaged 40 years of age, with 57.1% being married, and 52.2% having 1 to 2 children. Men averaged 36 years, with a marriage rate of 67%, and 47.8% having 1 to 2 children. Education levels and earnings varied slightly between sexes. Only increasing age (*p* < 0.001) and non-participation in a food assistance program (*p* = 0.04) were independently associated with the percentage contribution of UP to total energy intake, among the sociodemographic variables. These variables were included as covariates in the regression analyses of nutrient intake on UP consumption.

Table 2 illustrates energy and macronutrient consumption as percentages of the recommended daily intake amounts for individuals, across quintiles of the contribution of UP to total caloric intake. This result represents how caloric and macronutrient intake deviate from nutritional requirements across levels of increasing UP presence in the diet. Overall, the population showed deficits in meeting nutritional recommendations across most macronutrients. There were statistically significant positive associations (*p* < 0.0001) between higher UP consumption and energy, carbohydrate, total fat, saturated fats, monounsaturated fats, and trans-fat consumption, and omega-6 fatty acids (*p* = 0.033), while statistically significant negative associations were seen with protein (*p* = 0.037) and fiber (*p* = 0.035) intake.

Table 3 displays the mineral micronutrient intake as percentages of the recommend daily intake amounts, across quintiles of UP consumption. On average, the population demonstrated deficits in calcium, magnesium, and potassium intake, while exceeding recommended levels of manganese, phosphorus, iron, sodium, copper, selenium, and zinc. Notably, sodium consumption surpassed recommendations by 91.3%. Individuals with a higher UP intake tended to increase their consumption of calcium (*p* = 0.008) and manganese (*p* =< 0.001) and to decrease their intake of copper (*p* = 0.021).

Table 4 presents vitamin intake as a percentage of the recommend daily intake amounts, across quintiles of UP consumption. On average, the population fell short of recommended intakes for vitamins A, B6, D, and E, while exceeding recommendations for niacin, vitamin C, and B12. Individuals with a higher UP intake tended to consume more thiamine (*p* < 0.001) and vitamin B6 (*p* = 0.011), but less vitamin E (*p* = 0.002). There were also trends, close to statistical significance, towards a lower intake of vitamins A (*p* = 0.074) and B12 (*p* = 0.068).

## 4. Discussion

The study’s findings reveal that, when analyzing nutritional inadequacies according to recommended dietary guidelines, the overall population of workers exhibits deficits in energy and all macronutrients, as well as deviations in some micronutrients. The study showed that, with increasing UP contribution to the composition of the diet, there is a trend towards a higher energy intake. However, this heightened energy consumption primarily stems from increased fat intake, including saturated and trans fats, along with elevated carbohydrate intake and, conversely, decreased protein and fiber intake. This constitutes a distinctive nutritional profile of UP products, typically characterized by high energy density, rich in sugars, fats, and sodium, and deficient in fiber [7,13,14].

The results of our study are in line with the main results reported in the previously mentioned meta-analysis of 14 epidemiological studies based on national surveys [22]. Our study did not show an association of UP consumption with sodium intake, a relationship that has presented conflicting results in the literature. The meta-analysis showed that the greater consumption of UP did not contribute significantly to a higher sodium intake, but was positively correlated with the consumption of fats, and inversely with fiber, proteins and some micronutrients. Though our study and the meta-analysis did not show evidence of an association of UP consumption with sodium intake, other studies have reported both positive and negative associations. For example, a study based on data from the Korea National Health and Nutrition Examination Survey (KNHANES) between 2016 and 2018, with 16,657 adults and dietary intake assessed using a 1-day 24-hour recall, showed that the high energy contribution from UP was negatively associated with sodium intake [37].

Additionally, with increasing amounts of UP in the diet, there was also a statistically significant trend towards increased consumption of omega-6 fatty acids, commonly sourced from industrially processed oils [38], as well as selected minerals and vitamins, such as calcium, manganese, thiamine, and vitamin B6. In the opposite direction, increased consumption of UP was associated with a trend towards a lower intake of copper and vitamin E. As suggested by our results, other vitamin inadequacies may be associated with increased UP consumption, namely decreased intake of vitamins A and B12. The explanation for these dietary inadequacies may have been given by a study conducted in the United States (Seattle Obesity Study—SOS III) [39]. Involving over 800 adult participants, the study utilized a food frequency questionnaire and linear programming to generate nutritionally adequate dietary patterns, either combining unprocessed and UPs or analyzing them individually. The study revealed that in addition to UPs being major sources of added sugar, saturated fat, and sodium, they also contribute significantly to the intake of thiamine, niacin, and calcium. The study attributed this to UPs, such as white bread, being enriched with B-complex vitamins (e.g., thiamine, niacin, and riboflavin), while calcium was sourced from other foods, like yogurt, pizza, and fortified beverages. Actually, many UP products, including bread, ready-to-eat cereals, and certain beverages, are fortified with vitamins and minerals [39,40].

However, mitigating the energy deficit and some micronutrient deficiencies (through processed foods or fortification) observed in the population consuming more UP does not negate the detrimental effects of a diet high in fats, sodium, and low in protein, since this dietary profile is associated with the development of non-communicable diseases (NCDs), such as cardiovascular diseases, being overweight, obesity, and cancer, as has been extensively reported in many studies and reviews related to UP consumption [41,42,43]. Several mechanisms may be involved in the contribution of UPs to NCDs. On the one hand, the high energy density of fat and its presence in foods, or its addition in culinary preparations, is associated with greater palatability, which favors its consumption, especially in foods high in sugar, salt, and calories, and may lead to becoming overweight and obese [44]. On the other hand, UP consumption leads to a lower intake of less processed and more nutritious foods, like fruits, vegetables, whole grains, and minimally processed proteins, which are vital sources of proteins, fibers, vitamins, minerals, and other essential nutrients [7,16,41,45]. Moreover, a systematic review published as a consensus by the WHO, involving 84 studies, showed both a linear correlation between saturated fat consumption and plasma lipid concentrations, and an increase in serum total, LDL- and HDL-cholesterol concentrations with increasing amounts of UP in the diet [46].

Only a few studies have looked into nutritional inadequacies associated with increasing UP consumption, although all of them have reported the prevalence of subjects with nutrient intakes outside recommended quantities, differently from our study that quantified the amount of inadequate intake as percentage of the recommended amount for each subject, considering their age groups, caloric intake, and/or activity level. A British study with approximately 9000 participants found that the majority failed to meet the World Health Organization’s recommended intake for saturated fats, fibers, sodium, and potassium. As the contribution of UP to total energy intake increased, the prevalence of inadequate intakes of the studied nutrients significantly rose, notably sodium, which increased from 55.8% to 86.7% from the lowest to the highest quintile of UP consumption [16]. In Chile, a study involving 4920 individuals revealed that those in the highest quintile of UP consumption were significantly more likely to exceed dietary goals for trans-, total-, and saturated fats, and were also more likely to fall short of dietary goals for fiber density [47]. In Australia, a survey of over 12,000 individuals found that in the highest quintile of UP consumption, around 80% exceeded the recommended maximum limits for saturated fats and sodium, and over 85% failed to meet recommendations for fiber and potassium. Moreover, the prevalence of an inadequate intake increased from 6% to 11% for trans fats between the lowest and highest quintiles [48]. Studies were also conducted in France [49] and Portugal [50], and these presented similar results. While those studies did not analyze inadequacies in the intake of minerals and vitamins, they corroborate our findings regarding the increased intake of total, saturated, and trans fats associated with increased UP consumption.

Very few studies have studied the association between nutrient intake and UP consumption in specific categories of workers. A Spanish study on 1876 male workers in the automobile industry using a food frequency questionnaire showed a lower consumption of fiber and micronutrients (vitamins and minerals), such as vitamin B6, B9, B12, D, magnesium, and zinc [51]. Another cross-sectional study with 740 adult rural workers in Espirito Santo, southeastern Brazil, using three 24HR surveys, observed that greater UP consumption was associated with a higher caloric content of the diet and a lower content of all 23 macro- and micronutrients analyzed [52].

Nevertheless, it is important to note that nutritional recommendations are based on estimations. In this study, the midpoints of the ranges recommended by the aAcceptable Macronutrient Distribution Range (AMDR) were considered for macronutrient reference values, with saturated and trans fats limited to up to 10% and 1% of total energy consumed, respectively [36]. Adequate Intakes (AIs) were utilized for polyunsaturated fats (omega-3 and omega-6) when an average nutrient requirement could not be determined [29,53]. Recommended Dietary aAllowances (RDAs) or Adequate Intakes (AIs) were employed for micronutrients and, in the specific case of sodium, the chronic disease reference value from the Chronic Disease rRisk Reduction Intake (CDRRI) [33,34,35]. Despite their international recognition, these sources have limitations, such as not accounting for individual variations or dietary complexities.

Importantly, the population in this study comprises low-income workers, and the effects on nutrient intake associated with UP consumption may differ among non-low-income populations, a factor not extensively explored in studies. It is worth mentioning a study conducted in South Africa involving 2521 low-income adults aged 18 to 50 years [54], utilizing secondary data and 24HR data from cross-sectional studies, that assessed adherence to WHO dietary guidelines and UP consumption trends. It found that the likelihood of meeting international dietary guidelines is higher among those in the lowest quartile of UP consumption compared to those in the highest quartile. The study underscores the need for better strategies to protect low-income individuals, who are more susceptible to economic shocks resulting from poor health outcomes related to excessive UP consumption. It also highlights the prevalence of processed packaged foods on supermarket shelves and the greater shelf space allocated to unhealthy products than healthy ones [55,56], limiting consumers’ ability to make healthy food choices. Therefore, ensuring that healthy and nutritious foods are readily available, accessible, and appealing to consumers, including those with low incomes [54], could reduce deficiencies in macro- and micronutrients.

Micronutrient deficiencies pose a growing public health concern, and while fortification and food supplementation strategies have improved the status of certain micronutrients, effective alternative approaches are necessary to prevent these nutritional inadequacies. These may include investing in behavior change communication strategies, as it is essential to generate awareness and understand community perceptions of healthy eating [10,57]. Increasing awareness of dietary diversity and promoting the consumption of healthier food groups is a cost-effective way to enhance diet quality [58]. Policymakers should monitor UP consumption and implement appropriate measures to improve dietary habits. Furthermore, it is crucial for the industry to reformulate UPs to reduce fat, salt, and sugar content and to offer consumers minimally processed foods rich in micronutrients [10].

## 5. Limitations and Strengths

While this study provides new insights into the relationship of UPs to the macro- and micronutrient composition of diets, it has limitations, such as using nutritional recommendations based on estimations from population studies, which may not fully capture the reality of the sample individuals. However, these are the prevailing references in the literature used in clinical practice and scientific research. To evaluate food consumption, the 24HR method was used instead of other methods, such as the Food Frequency Questionnaire (FFQ). Our choice is due to the fact that the FFQ identifies the foods consumed and their frequency of consumption without detailing the quantities consumed, whereas the 24HR method allows for a detailed description of food consumption, detailing ingredients and quantities. This method can be used to quantify and characterize food consumption, either by food groups, by calories, or by nutrients, which aligns with the objectives of our study. Furthermore, to reduce possible memory biases among interviewees, the multiple pass methods methodology was used, and the survey was administered by previously trained nutritionists. Although the average of two or more applications of the 24HR method should capture more accurately the usual food consumption, it is common practice to apply it only once. In the abovementioned meta-analysis of 14 studies [22], the one-day 24HR method was used in half the studies, and only three used the multiple pass methodology. An additional limitation is that the research was conducted with a sample from a single federation state, although there are no significant socioeconomic and biodemographic differences compared to other states in the northeast region of the country. On the positive side, to the best of our knowledge, this study is the first to analyze nutritional inadequacies based on deviations in nutrient intake according to nutritional recommendations concerning UP consumption quintiles. Other strengths of the study methodology are a large probability sample, representative of manufacturing industries of an entire state, data collection conducted prospectively and personally by trained nutritionists, and the absence of non-respondents.

## 6. Conclusions

This study on a representative sample of manufacturing workers from northeastern Brazil showed consumption below dietary reference intakes for energy, all macronutrients, and some micronutrients. With the increasing contribution of UPs to the composition of the diet, there is a reduction in energy deficiency, but this increase in energy occurs mainly due to the increase in fat intake, including saturated and trans fats, together with a high carbohydrate intake, with a negative consequence of reduced protein consumption. Correcting energy and micronutrient deficiencies by increasing the consumption of UPs may lead to a decrease in diet quality. Dietary inadequacies were also seen in several minerals and vitamins.

## Figures and Tables

**Table 1 ijerph-21-00888-t001:** Sociodemographic characteristics of the study population.

Variables	Total (*n* = 921)	Females (*n* = 406)	Males (*n* = 515)
Age (years)	38.2 ± 10.7	40.0 ± 9.69	36.8 ± 11.3
Civil status			
Single/divorced/widowed	344 (37.4%)	174 (42.9%)	170 (33.0%)
Married/living with partner	577 (62.7%)	232 (57.1%)	345 (67.0%)
Number of children			
None	283 (30.7%)	110 (27.1%)	173 (33.6%)
1–2	458 (49.7%)	212 (52.2%)	246 (47.8%)
3+	180 (19.5%)	84 (20.7%)	
Education			
Elementary school	159 (17.3%)	50 (12.3%)	109 (21.2%)
Middle school	235 (25.5%)	107 (26.4%)	128 (24.9%)
High school	430 (46.7%)	203 (50.0%)	227 (44.1%)
Higher education	97 (10.5%)	46 (11.3%)	51 (9.90%)
Income (minimum wages)	1.46 ± 1.61	1.22 ± 0.62	1.64 ± 2.07
In-house training	172 (18.7%)	69 (17.0%)	103 (18.7%)

Tabulated values are the mean ± standard deviation or *n* (%).

**Table 2 ijerph-21-00888-t002:** Macronutrient intake (percentage of recommended amounts) across quintiles of the contribution of ultra-processed food consumption to total caloric intake.

Nutrient	Overall		Quintiles of % Contribution of Ultra-Processed Foods to Total Energy Intake					*p*-Value Trend
	Mean	SD	Q1	Q2	Q3	Q4	Q5	
Energy intake ^c^	−11.2	32.1	−20.2	−14.3	−7.86	−8.73	−4.91	**<0.001**
Proteins ^a^	−23.1	30.7	−22.6	−21.7	−19.0	−25.2	−27.0	**0.037**
Carbohydrates ^a^	−15.5	32.5	−24.4	−17.6	−13.2	−12.5	−10.1	**<0.001**
Total lipid ^a^	−9.13	45.1	−24.4	−17.4	−4.93	−5.11	6.18	**<0.001**
Saturated fats	−16.3	49.7	−30.2	−23.0	−12.8	−13.1	−2.31	**<0.001**
Polyunsaturated fatty acids ^a^	−23.3	30.0	−25.5	−24.4	−22.1	−20.2	−24.4	0.98
Monounsaturated fatty acids ^a^	−17.3	43.1	−30.3	−25.1	−13.0	−13.7	−4.33	**<0.001**
n6-polyunsaturated fatty acids ^b^	−23.8	46.3	−32.9	−28.9	−18.9	−17.6	−20.6	**0.033**
n3-polyunsaturated fatty acids ^b^	−11.1	56.6	−16.9	−12.8	−4.92	−7.69	−13.0	0.63
Trans fatty acids	−8.79	66.6	−31.7	−25.8	−15.1	10.9	17.8	**<0.001**
Dietary fiber	−12.2	40.7	−10.3	−12.6	−9.14	−12.4	−16.7	**0.035**

Sources: ^a^—Acceptable Macronutrient Distribution Range (AMDR); ^b^—Adequate Intake (AI); ^c^—Estimated Energy Requirement (EER). Statistically significant p-Values are in bold. Nonparametric multiple linear regression adjusted by age and participation in a food assistance program.

**Table 3 ijerph-21-00888-t003:** Micronutrient intake (percentage of recommended amounts) across quintiles of the contribution of ultra-processed food consumption to total caloric intake.

Nutrient	Overall		Quintiles of % Contribution of Ultra-Processed Foods to Total Energy Intake					*p*-Value Trend
	Mean	SD	Q1	Q2	Q3	Q4	Q5	
Calcium ^a^	−54.5	39.6	−61.2	−56.1	−54.4	−51.1	−49.8	**0.008**
Magnesium ^a^	−36.2	23.9	−38.8	−39.7	−33.9	−35.2	−33.2	0.095
Manganese ^c^	15.0	77.3	3.10	−0.96	9.32	26.5	37.2	**<0.001**
Phosphorus ^a^	69.5	76.9	64.7	68.2	73.7	71.5	69.1	0.81
Iron ^a^	1.48	68.0	−2.38	−3.08	0.20	4.04	8.62	0.11
Sodium ^b^	91.2	90.9	82.8	83.7	97.5	94.0	98.0	0.51
Potassium ^c^	−24.0	25.9	−25.4	−24.7	−21.3	−24.0	−24.4	0.70
Copper ^a^	40.8	309.3	75.7	63.2	30.0	22.5	12.7	**0.033**
Selenium ^a^	48.9	100.9	47.5	38.7	56.5	47.2	54.3	0.72
Zinc ^a^	16.8	62.9	16.9	14.8	20.1	20.3	11.8	0.074

Sources: ^a^—Recommended Dietary Allowance (RDA); ^b^—Chronic Disease Risk Reduction Intake (CDRRI); ^c^—Adequate Intake (AI). Statistically significant p-Values are in bold. Nonparametric multiple linear regression adjusted by age and participation in a food assistance program.

**Table 4 ijerph-21-00888-t004:** Vitamin intake (percentage of recommended amounts) across quintiles of the contribution of ultra-processed food consumption to total caloric intake.

Nutrient	Overall		Quintiles of % Contribution of Ultra-Processed Foods to Total Energy Intake					*p*-Value Trend
	Mean	SD	Q1	Q2	Q3	Q4	Q5	
Vitamin A ^a^	−19.1	354.8	10.2	18.6	−38.7	−35.1	−50.5	0.074
Thiamin ^a^	−3.23	60.7	−24.3	−17.8	−3.40	6.49	22.8	**<0.001**
Riboflavin ^a^	−3.83	71.8	−10.8	−5.43	−2.85	0.49	−0.53	0.64
Vitamin B6 ^a^	−28.0	65.3	−36.5	−31.9	−23.3	−20.1	−28.3	**0.011**
Niacin ^a^	28.6	78.0	21.3	28.8	25.0	35.7	32.2	0.25
Vitamin C ^a^	63.8	296.8	67.1	96.5	35.4	64.3	53.8	0.41
Vitamin D ^a^	−82.4	26.4	−84.0	−79.8	−83.7	−81.8	−83.7	0.27
Vitamin E ^a^	−70.5	16.0	−69.2	−70.3	−70.0	−70.8	−72.1	**0.002**
Vitamin B12 ^a^	138.9	529.7	165.3	189.7	145.6	119.7	74.2	0.068

Sources: ^a^—Recommended Dietary Allowance (RDA). Statistically significant p-Values are in bold. Nonparametric multiple linear regression adjusted by age and participation in a food assistance program.

## Data Availability

Study data will be available upon reasonable request to the corresponding author.

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
