# Peer review of "Quantitative Assessment of the Inadequate Intake of Macronutrients, Minerals, and Vitamins Associated with Ultra-Processed Food Consumption"

_ijerph, 2024, doi:10.3390/ijerph21070888_

Round 1
Reviewer 1 Report
Comments and Suggestions for Authors
This study examined the relationship between ultra-processed food intake and energy and nutrient intake in manufacturing workers and is important at a time when the high intake of ultra-processed foods is becoming an issue.
There are several suggestions for improving this study.
It would be easier to understand the gender differences for each item in Table 1 by showing P-values.
Some items (e.g. age, educational history) are likely to show significant differences. This may mean that this population cannot be considered as one. In other words, characteristics such as gender and educational history need to be taken into account when examining the association between processed food intake and nutrient intake.
For example, I would recommend a stratified analysis by sex, or adjusting for sex and other factors in a multiple regression analysis.
You should include more discussion of the results of this study of manufacturing workers and the results of studies examining the relationship between ultra-processed food intake and nutrient intake in other industries.
In this way, the realities of manufacturing workers will be better understood and nutrition policy will be better informed in the future.
Author Response
We would like to thank the reviewer for the valuable suggestions, which were of great importance in improving our text. We appreciate your useful and pertinent comments, which we respond to and accept as described below:
Comment: It would be easier to understand the gender differences for each item in Table 1 by showing P-values.
Response: Thank you for your comments. The reason why we presented the study population characteristics disaggregated by sample was to support the Sex and Gender Equity in Research (SAGER) guidelines, that recommend the presentation of research data and results separately by gender, when appropriate. We prefer not to present p-values because it might give the wrong impression that results were to be compared between sexes. We did not compare associations of nutrient intake inadequacies with UP consumption between sexes because we saw no reason why such relationships could be influenced by gender.
Comment: Some items (e.g. age, educational history) are likely to show significant differences. This may mean that this population cannot be considered as one. In other words, characteristics such as gender and educational history need to be taken into account when examining the association between processed food intake and nutrient intake.
For example, I would recommend a stratified analysis by sex, or adjusting for sex and other factors in a multiple regression analysis.
Response: Although we prefer the analysis we have presented in the initial version of the manuscript, we understand the reviewer’s comment and we decided to comply with the request, at the cost of a few changes in the results. Our original analysis tested the null hypothesis that no trend existed in nutrient inadequate intake across quintiles of % contribution of UP consumption to total energy intake, with a nonparametric test. The advantage of this approach is that the test would be significant whether the trend was linear or non-linear, while multiple regression will not identify non-linear trends. However, the nonparametric test we used did not allow adjustment for confounders, which may be considered an important limitation. Therefore, we changed the statistical methodology so that adjustment for covariates was possible. As associations between UP consumption and nutrient intake may be nonlinear, as the distribution of most nutrients is highly skewed and therefore non-normal, multiple linear regression would not be adequate. Therefore, we use nonparametric regression, which overcomes both limitations of linear regression. In all analyses, demographic variables that have shown in multiple linear regression to be independently associated with % contribution of UP consumption to total energy intake were included as covariates. In the Tables, we have maintained the display of the descriptive results over quintiles of UP consumption for easier appreciation of the trends. Therefore, the statistical analysis section was amended as follows (lines 162-172):
For the investigation of the association of UP consumption with nutrient inadequacy, a search for potential confounders was done with stepwise multiple linear regression of the sociodemographic variables (age, sex, marriage status, number of children, education, income, in-house training, and participation in a food assistance program) on the percent contribution of UP consumption to total energy intake. The variables selected by backward elimination (p<0.05) were included as covariates in all statistical analyses. Because of the possibility of non-linear associations between UP consumption and nutrient inadequate intake, as well as the highly skewed distribution of most nutrients, nonparametric multiple regression was used to test the association of nutrient intake with UP consumption.
When redoing the analysis, we identified a worker with incorrect data. These data were corrected.
Comment: You should include more discussion of the results of this study of manufacturing workers and the results of studies examining the relationship between ultra-processed food intake and nutrient intake in other industries. In this way, the realities of manufacturing workers will be better understood and nutrition policy will be better informed in the future.
Response: To attend your suggestion, the following paragraph was added (lines 315-322):
Very few studies have studied the association between on nutrient intake and UP consumption in specific categories of workers. A Spanish study on 1876 male workers in the automobile industry using a food frequency questionnaire showed lower consumption of fiber and micronutrients (vitamins and minerals) such as vitamin B6, B9, B12, D, magnesium, and zinc [51]. Another cross-sectional study with 740 adult rural workers in Espirito Santo, southeastern Brazil, using three 24HR observed that greater UP consumption was associated with higher caloric content of the diet and lower content of all 23 macro- and micronutrients analyzed [52].
Reviewer 2 Report
Comments and Suggestions for Authors
In this study authors evaluated the Interaction between Nutrient Intake and Ultra- processed Food Consumption; however I have few suggestions to improve the manuscript.
1. Abstract: Please mention the method used to evaluate nutrient intake of the population. Also add one concluding remark in the end of the abstract.
2. Title: title is too long. It is better to change the title as
a. Interaction between Nutrient Intake and Ultra-2 processed Food Consumption.
or
b. Effect of Ultra-processed Food Consumption on Macronutrient, Mineral and Vitamin Inadequacies
3. Introduction: Last paragraph should be revised and improved through adding previous studies on this topic, research gaps and importance of the current study. Add a few more studies and relevant literature.
4. Discussion: Please add a mechanistic approach regarding the effects of high fat intake on biological mechanisms and adverse health outcomes.
5. Limitation: Add a limitation section at the end of the manuscript discussing strengths and limitations of the study. I am wondering that authors used 24 hours recall for analysis of the interaction between UP food and nutrients intake of the population. However, only 24 hours recall is not enough to measure habitual intake. FFQ along with 24 hours recall can be a better approach to analyze habitual intake. This should be discussed in the limitation section.
Author Response
We would like to thank the reviewer for the valuable suggestions, which were of great importance in improving our text. We appreciate your useful and pertinent comments, which we respond to and accept as described below:
In this study authors evaluated the Interaction between Nutrient Intake and Ultra- processed Food Consumption; however, I have few suggestions to improve the manuscript.
Comment: Abstract: Please mention the method used to evaluate nutrient intake of the population. Also add one concluding remark in the end of the abstract.
Response: Thank you for your comments. To clarify the method used to evaluate nutrient intake of the population were added at abstract (lines 19-20)
“Food consumption was extracted from the 24-hour recall survey..”
And, to concluding remark in the end of the abstract, we added (lines 31-32)
“The results show that correcting energy and micronutrient deficiencies by increasing UP consumption can also lead to a decrease in diet quality.”
Comment: Title: title is too long. It is better to change the title as
- Interaction between Nutrient Intake and Ultra-2 processed Food Consumption.
or
- Effect of Ultra-processed Food Consumption on Macronutrient, Mineral and Vitamin Inadequacies
Response: To reduce the length of the article title, as suggested by the reviewer, we changed it to the following shorter version:
Quantitative Assessment of Inadequate Intake of Macronutrients, Minerals and Vitamins associated with Ultra-processed Foods Consumption.
Comment: Introduction: Last paragraph should be revised and improved through adding previous studies on this topic, research gaps and importance of the current study. Add a few more studies and relevant literature.
Response: To add some more studies and relevant literature and improve the introduction, we added the following paragraph (lines 60-64)
Several studies that investigated the association between the consumption of UP foods and nutrient content of diets have consistently found a trend towards increased amounts of energy, carbohydrates, total and saturated fat in the composition of diets, with simultaneous decrease of protein and fiber, as well as a number of minerals and vitamins [22].
And to evidence the research gaps and importance of the current study, we add the following paragraph (lines 64-70):
All those studies have reported only the average consumed amounts of nutrients across categories of UP consumption, with a smaller number also estimating the proportion of subjects consuming inadequate amounts of nutrients, with cut-off levels usually based on tables of Estimated Average Requirements or Tolerable Upper Intake Levels. However, no study has yet, to the best of our knowledge, estimated the magnitude of the deviations from Recommended Dietary Allowances associated with levels of UP consumption.
Comment: Discussion: Please add a mechanistic approach regarding the effects of high fat intake on biological mechanisms and adverse health outcomes.
Response: In order to explain better that mechanistic approach,
the following paragraph (lines 225-231)
However, mitigating the energy deficit and some micronutrient deficiencies (through processed foods or fortification) observed in the population consuming more UP does not negate the detrimental effects of a diet high in fats, sodium, and low in protein. UP consumption leads to lower intake of less processed and more nutritious foods like fruits, vegetables, whole grains, and minimally processed proteins, which are vital sources of proteins, fibers, vitamins, minerals, and other essential nutrients [7,16,41,42], potentially resulting in reduced consumption of these components in the general population.
Was changed to (lines 274-291)
However, mitigating the energy deficit and some micronutrient deficiencies (through processed foods or fortification) observed in the population consuming more UP does not negate the detrimental effects of a diet high in fats, sodium, and low in protein, since this dietary profile is associated with the development of Non Communicable Diseases (NCDs) such as cardiovascular diseases, overweight, obesity and cancer, as has been extensively reported in many studies and reviews related to UP consumption [41-43]. Several mechanisms may be involved in the contribution of UP foods to NCDs. On the one hand, the high energy density of fat and its presence in foods, or its addition in culinary preparations, is associated with greater palatability, which favors its consumption, especially in foods high in sugar, salt and calories, and may lead to overweight and obesity [44]. On the other hand, UP consumption leads to lower intake of less processed and more nutritious foods like fruits, vegetables, whole grains, and minimally processed proteins, which are vital sources of proteins, fibers, vitamins, minerals, and other essential nutrients [7,16,41,45]. Moreover, a systematic review published as a consensus by the WHO involving 84 studies, showed both a linear correlation between saturated fat consumption and plasma lipid concentrations, and an increase in serum total, LDL- and HDL-cholesterol concentrations with increasing amounts of UP in the diet [46].
Comment: Limitation: Add a limitation section at the end of the manuscript discussing strengths and limitations of the study. I am wondering that authors used 24 hours recall for analysis of the interaction between UP food and nutrients intake of the population. However, only 24 hours recall is not enough to measure habitual intake. FFQ along with 24 hours recall can be a better approach to analyze habitual intake. This should be discussed in the limitation section.
Response: to attend your suggestion, was added a section “5. Limitations and strengths” (line XX) and to make clear about the use of the 24 hours recall,
the following paragraph (lines 289-301):
While this study provides new insights into the relationship of UP to the macro- and micronutrient composition of diets, it has limitations, such as using nutritional recommendations based on estimations from population studies, which may not fully capture the reality of the sample individuals. However, these are the prevailing references in the literature used in clinical practice and scientific research. Additionally, the research was conducted with a sample from a single federation state, although there are no significant socioeconomic and biodemographic differences compared to other states in the northeast region of the country. On the positive side, to the best of our knowledge, this study is the first to analyze nutritional inadequacies based on deviations in nutrient intake according to nutritional recommendations concerning UP consumption quintiles. Other strengths of the study methodology are a large probability sample, representative of manufacturing industries of an entire state, data collection conducted prospectively and personally by trained nutritionists, and absence of non-respondents.
Was changed to (lines 362-389):
While this study provides new insights into the relationship of UP to the macro- and micronutrient composition of diets, it has limitations, such as using nutritional recommendations based on estimations from population studies, which may not fully capture the reality of the sample individuals. However, these are the prevailing references in the literature used in clinical practice and scientific research. To evaluate food consumption, the 24-hour dietary recall was used instead of other methods such as the Food Frequency Questionnaire (FFQ). Our choice is due to the fact that the FFQ identifies the foods consumed and their frequency of consumption without detailing the quantities consumed, whereas the 24-hour recall allows for a detailed description of food consumption, detailing ingredients and quantities. This method can be used to quantify and characterize food consumption, either by food groups, by calories, and by nutrients, which aligns with the objectives of our study. Furthermore, to reduce possible memory biases among interviewees, the Multiple Pass Methods methodology was used and the survey being administered by previously trained nutritionists. Although the average of two or more applications of the 24HR should capture more accurately the usual food consumption, it is common practice to apply only once. In the previously mentioned meta-analysis of 14 studies [22], the one-day 24HR was used in half the studies and only three used the Multiple Pass Methodology. An additional limitation is that the research was conducted with a sample from a single federation state, although there are no significant socioeconomic and biodemographic differences compared to other states in the northeast region of the country. On the positive side, to the best of our knowledge, this study is the first to analyze nutritional inadequacies based on deviations in nutrient intake according to nutritional recommendations concerning UP consumption quintiles. Other strengths of the study methodology are a large probability sample, representative of manufacturing industries of an entire state, data collection conducted prospectively and personally by trained nutritionists, and absence of non-respondents.
Round 2
Reviewer 1 Report
Comments and Suggestions for Authors I checked your all comments, and agreed with you revise.